# Vaccines to prevent COVID-19: A living systematic review with Trial Sequential Analysis and network meta-analysis of randomized clinical trials

Steven Kwasi Korang[1]*, Elena von Rohden[1], Areti Angeliki Veroniki[2,3], Giok Ong[4], Owen Ngalamika[5], Faiza Siddiqui[1], Sophie Juul[1], Emil Eik Nielsen[1], Joshua Buron Feinberg[1], Johanne Juul Petersen[1], Christian Legart[1,6], Afoke Kokogho[7], Mathias Maagaard[1,8], Sarah Klingenberg[1,9], Lehana Thabane[10], Ariel Bardach[11], Agustín Ciapponi[11], Allan Randrup Thomsen[12], Janus C. Jakobsen[1,9,13], Christian Gluud[1,9,13]

1 Copenhagen Trial Unit, Centre for Clinical Intervention Research, The Capital Region, Copenhagen University Hospital – Rigshospitalet, Copenhagen, Denmark, 2 Knowledge Translation Program, Li Ka Shing Knowledge Institute, St. Michael's Hospital, Toronto, Ontario, Canada, 3 Department of Metabolism, Digestion and Reproduction & Department of Surgery and Cancer, Faculty of Medicine, Imperial College London, United Kingdom, 4 Systematic Review Initiative, NHS Blood and Transplant, John Radcliffe Hospital, Headley Way, Oxford, United Kingdom, 5 Dermatology & Venereology Division, University Teaching Hospital, University of Zambia School of Medicine, Lusaka, Zambia, 6 Center for Clinical Metabolic Research, Gentofte Hospital, University of Copenhagen, Hellerup, Denmark, 7 United States Army Medical Research Directorate West Africa, Henry M. Jackson Foundation Medical Research International (HJFMRI), Walter Reed Army Institute of Research, Abuja, Nigeria, 8 Centre for Anaesthesiological Research, Department of Anaesthesiology, Zealand University Hospital, The Zealand Region of Denmark, Køge, Denmark, 9 The Cochrane Hepato-Biliary Group, Copenhagen Trial Unit, Centre for Clinical Intervention Research, The Capital Region, Copenhagen University Hospital – Rigshospitalet, Copenhagen, Denmark, 10 Department of Health Research Methods, Evidence, and Impact, McMaster University, Hamilton, Ontario, Canada, 11 Argentine Cochrane Center. Instituto de Efectividad Clínica y Sanitaria (IECS-CONICET), Buenos Aires, Argentina, 12 Department of Immunology and Microbiology, University of Copenhagen, Copenhagen, Denmark, 13 Department of Regional Health Research, The Faculty of Heath Sciences, University of Southern Denmark, Odense, Denmark

* steven.korang@ctu.dk

**Data Availability Statement:** All relevant data are within the manuscript and its Supporting information files.

## Abstract

### Background

COVID-19 is rapidly spreading causing extensive burdens across the world. Effective vaccines to prevent COVID-19 are urgently needed.

### Methods and findings

Our objective was to assess the effectiveness and safety of COVID-19 vaccines through analyses of all currently available randomized clinical trials. We searched the databases CENTRAL, MEDLINE, Embase, and other sources from inception to June 17, 2021 for randomized clinical trials assessing vaccines for COVID-19. At least two independent reviewers screened studies, extracted data, and assessed risks of bias. We conducted meta-analyses, network meta-analyses, and Trial Sequential Analyses (TSA). Our primary outcomes

**Funding:** The Copenhagen Trial Unit provided support in the form of salaries for SKK, EvR, FS, SJ, EEN, JF, JJP, JCJ and CG. The specific roles of these authors are articulated in the 'author contributions' section.

**Competing interests:** The authors have declared that no competing interests exist.

included all-cause mortality, vaccine efficacy, and serious adverse events. We assessed the certainty of evidence with GRADE. We identified 46 trials; 35 trials randomizing 219 864 participants could be included in our analyses. Our meta-analyses showed that mRNA vaccines (efficacy, 95% [95% confidence interval (CI), 92% to 97%]; 71 514 participants; 3 trials; moderate certainty); inactivated vaccines (efficacy, 61% [95% CI, 52% to 68%]; 48 029 participants; 3 trials; moderate certainty); protein subunit vaccines (efficacy, 77% [95% CI, −5% to 95%]; 17 737 participants; 2 trials; low certainty); and viral vector vaccines (efficacy 68% [95% CI, 61% to 74%]; 71 401 participants; 5 trials; low certainty) prevented COVID-19. Viral vector vaccines decreased mortality (risk ratio, 0.25 [95% CI 0.09 to 0.67]; 67 563 participants; 3 trials, low certainty), but comparable data on inactivated, mRNA, and protein subunit vaccines were imprecise. None of the vaccines showed evidence of a difference on serious adverse events, but observational evidence suggested rare serious adverse events. All the vaccines increased the risk of non-serious adverse events.

## Conclusions

The evidence suggests that all the included vaccines are effective in preventing COVID-19. The mRNA vaccines seem most effective in preventing COVID-19, but viral vector vaccines seem most effective in reducing mortality. Further trials and longer follow-up are necessary to provide better insight into the safety profile of these vaccines.

## Introduction

COVID-19 is caused by Severe Acute Respiratory Syndrome Coronavirus 2 (SARS-Cov-2) [1]. Since the observation of the first patients with COVID-19 in Wuhan in 2019, the disease quickly became a global pandemic [2]. As of September 6th, 2021 more than 221 million individuals developed COVID-19 and 4.6 million died globally [2].

Currently, there are only a few interventions hat seem able to benefit the clinical course of COVID-19 [3]. Therefore, preventive measures are of vital importance to control COVID-19. The scientific, medical, and industrial communities have embarked on efforts to develop safe vaccines [4, 5]. At September 13th, 2021 about 104 vaccines are being tested in humans, with 33 having reached phase III trials [6]. Of these, five vaccines were abandoned due to either lack of response or an undesirable adverse event (e.g., tested false positive for HIV), eight have been approved for full use, and eleven have been authorized for early or limited use [6]. With the rapid development, approval of vaccines for COVID-19, and the growing number of viral variants, there is a need for systematic reviews critically appraising the topic. We recently published the first version of our review as a preprint, showing effects of mRNA and viral vector vaccines [7].

The aim of this second version of our living systematic review is to assess the effectiveness and safety of COVID-19 vaccines through analyses of all currently available randomized clinical trials. We also narratively describe incidentally identified observational studies reporting harms that we encountered during our search for trials [8].

## Methods

This systematic review with meta-analyses was conducted in accordance with the reporting guideline provided in the Preferred Reporting Items for Systematic Reviews and Meta-

Analysis (PRISMA) statement [9, 10]. This review was carried out following recommendations outlined in the Cochrane Handbook of Systematic Reviews of Interventions [11]. More details on our methods can be found in our published protocol [8, 12].

## Search strategy and selection criteria

An experienced information specialist performed weekly literature searches. For details regarding databases, searches, and screening methods see our published protocol [8] or S1 File.

We searched for and included randomized clinical trials, irrespective of publication status, year, and language. We included trials with any participant irrespective of prior exposure, age, sex, comorbidities, immune status, and risk group [8].

We included any vaccine aiming to prevent COVID-19 irrespective of dose and duration of administration. We included randomized clinical trials with any control group, i.e. vaccine versus placebo, 'active placebo', no intervention, another vaccine aiming at preventing COVID-19, or any other 'active' comparator [8].

## Data analysis

Two authors independently screened and extracted data. Our primary outcomes were all-cause mortality; vaccine efficacy defined by either preventing COVID-19 symptoms plus positive polymerase chain rection (PCR) test, preventing severe COVID-19 symptoms plus positive PCR test, or preventing positive PCR test only; and serious adverse events [10]. Our secondary outcomes were health-related quality of life and adverse events not considered serious [8]. We used the trial results reported at maximum follow-up for all outcomes. We used intention-to-treat data if provided by the trialists [8]. Several exploratory outcomes were predefined in our protocol and will be presented in a subsequent publication [8].

**Risk of bias assessment.** Our risk of bias assessment was based on the Cochrane risk of bias tool version 2 (RoB 2) [13].

**Association measures.** We calculated risk ratios (RRs) with 95% confidence interval (CI). In meta-analysis, we also calculated the Trial Sequential Analysis-adjusted CIs. We used forest plots to illustrate summary effect sizes of the comparative effectiveness among interventions.

**Statistical synthesis.** We undertook our analysis according to the Cochrane Handbook of Systematic Reviews of Interventions [11], Keus and colleagues [14], and our eight-step assessment [15]. We used Stata version 16.1 to analyse data using the *metan* command for meta-analysis and *network* suite of commands for network meta-analysis [16–18]. We assessed our intervention effects with both a random-effects (RE) meta-analysis (DerSimonian and Laird) [19] and a fixed-effect (FE) meta-analysis (Mantel-Haenszel) [20] for each preventive comparison. We reported the more conservative point estimate of the two [8, 15]. We assessed three primary outcomes and two secondary outcomes. We, therefore, considered a p value of 0.0167 or less as the threshold for statistical significance to adjust for multiplicity [8, 15].

We performed Trial Sequential Analysis on all outcomes, to calculate the diversity-adjusted required information size (DARIS) and the cumulative Z-curve's breach of relevant trial sequential monitoring boundaries [21–23].

We performed a network meta-analysis as described in S1 File and our protocol [8].

We assessed heterogeneity through visual inspection of forest plots, using the $I^2$ statistic [11, 24, 25], and by estimating the between-study variance using the DerSimonian and Laird method [26, 27]. We investigated heterogeneity through subgroup analyses.

**Additional analyses.** We performed subgroup analysis assessing different types of vaccines; trials at low compared with at high risk of bias; and trials without against with vested

interests. To assess the potential impact of missing data, we performed sensitivity analyses using 'best-worst' and 'worst-best' analysis [8, 15].

**Summary of findings and assessment of certainty.**   We created summary of findings tables including each of the prespecified outcomes (all-cause mortality, vaccine efficacy, serious adverse events, health-related quality of life, and non-serious adverse events) (Tables 1–4). We used the five Grading of Recommendations, Assessment, Development, and Evaluation (GRADE) considerations (bias risk of the trials, consistency of effect, imprecision, indirectness, and publication bias) and CINeMA to assess confidence in the meta-analysis and network meta-analysis findings, respectively [28–31]. We assessed imprecision using Trial Sequential Analysis [8, 32, 33].

## Results

46 trials met our inclusion criteria [34–65], of which 35 trials [34, 36–42, 44–50, 53–59, 61–65] randomizing 219 864 participants provided data for our predefined meta-analyses. See PRISMA flowchart (Fig 1) for details regarding the literature search and the selection of trials.

In most of the trials reporting data for our meta-analysis, the included participants were adults. Two of the trials included children (<16 years) [55, 64], but none of the trials included pregnant women or immunocompromised participants (e.g. cancer; on cancer drugs; etc.). All included trials assessed a COVID-19 vaccine compared with placebo (n = 32) or a control vaccine not immunogenic towards COVID-19 (n = 3) [37, 45]. The included vaccines were based on mRNA [34, 41, 42, 66]; viral vectors [37, 39, 43, 45, 47, 50, 52, 67]; protein-subunit [38, 53]; or inactivated virus [47–49, 51] (Table 5). The median follow-up range of assessment varied from 35 to 92 days after randomization for all outcomes.

Ten trials were at overall low risk of bias [43, 47, 48, 67], seventeen trials were at overall some concerns [34, 36, 38, 39, 41, 42, 46, 51–53], and eight trials were at overall high risk of bias (S1 File) [37, 40, 45].

### All-cause mortality

**Inactivated vaccines.**   Six trials assessing inactivated vaccines reported on all-cause mortality. Meta-analysis (FE) showed that these vaccines versus controls may result in a large reduction of all-cause mortality, but the confidence interval was compatible with no effect (risk ratio (RR), 0.48 [95% CI 0.12 to 1.97]; p = 0.31; $I^2$ = 0.0%; 53 399 participants; very low certainty; Fig 2).

**mRNA vaccines.**   Five trials assessing mRNA vaccines reported on all-cause mortality. Meta-analysis (RE) showed that these vaccines versus placebo may reduce all-cause mortality, but the confidence interval was compatible with no effect (RR, 0.63 [95% CI 0.21 to 1.84]; p = 0.39; $I^2$ = 0%; 75 926 participants; low certainty; Fig 2).

**Protein-subunit vaccines.**   Four trials assessing protein-subunit vaccines reported on all-cause mortality. Meta-analysis (FE) showed that these vaccines versus controls may result in a reduction of all-cause mortality, but the confidence interval was compatible with no effect (RR, 0.46 [95% CI 0.09 to 2.36]; p = 0.35; $I^2$ = 0.0%; 15 634 participants; very low certainty; Fig 2).

**Viral vector vaccines.**   Three trials assessing viral vector vaccines reported on all-cause mortality. Meta-analysis (FE) showed that these vaccines versus controls may result in a large reduction of all-cause mortality (RR, 0.25 [95% CI 0.09 to 0.67]; p = 0.01; $I^2$ = 0.0%; 67 563 participants; low certainty; Fig 2).

**Trial sequential analysis and sensitivity analysis on mortality.**   Trial Sequential Analyses showed that we did not have enough information to confirm that any of the vaccines reduced

**Table 1. Summary of findings (inactivated vaccines).**

Inactivated vaccines versus placebo

**Population**: General population
**Settings**: Outpatient
**Intervention**: Inactivated vaccine (BBIBP-CorV, CoronaVac, Covaxin, and Vero Cell)
**Comparison**: Placebo

| Outcomes | Illustrative comparative risks* (95% CI) | | Relative effect (95% CI) | No of participants (studies) | Quality of the evidence (GRADE) | Comments |
|---|---|---|---|---|---|---|
| | Assumed risk | Corresponding risk | | | | |
| | Control | Vaccine | | | | |
| **All-cause mortality** maximum follow-up | **Study population** | | 0.48 (0.12 to 1.97) | 53 399 | ⊕⊖⊖⊖ **Very low** | Downgraded one level for serious risk of bias and two levels for very serious imprecision. DARIS: 1 350 077 (Pc 0.03%; RRR 20%; alpha 1.67%; beta 10%; diversity 0%) |
| | **10 per 100,000** | **4 per 100,000** (1 to 20) | | | | |
| **Vaccine efficacy** Positive test plus symptoms maximum follow-up | **Study population** | | 61% (52 to 68%) | 48 029 (3) | ⊕⊕⊕⊖ **Moderate** | Downgraded one level for serious risk of bias DARIS: 69 896 (Pc 1.5%; RRR 50%; alpha 1.67%; beta 10%; diversity 84.92%) |
| | **152 per 10,000** | **59 per 10,000** (49 to 73) | | | | |
| **Serious adverse events** maximum follow-up | | | 0.84 (0.68 to 1.06) | 53 839 (7) | ⊕⊕⊖⊖ **Low** | Downgraded one level for serious risk of bias and one level for serious imprecision. DARIS: 373 676 (Pc 0.55%; RRR 20%; alpha 1.67%; beta 10%; diversity 41.4%) |
| | **55 per 10,000** | **47 per 10,000** (38 to 59) | | | | |
| **Health-related quality of life** maximum follow-up | | | NA | NA | NA | No trials assessed health-related quality of life |
| | NA | NA | | | | |
| **Non- serious adverse events** maximum follow-up | **5733 per 10,000** | **5848 per 10,000** (5274 to 6478) | 1.02 (0.92 to 1.13) | 54 239 (11) | ⊕⊕⊕⊖ **Moderate** | Downgraded one level for serious risk of bias DARIS: 63 233 (Pc 57.3%; RRR 20%; alpha 1.67%; beta 10%; diversity 98.4%) |

*The basis for the **assumed risk** (e.g., the median control group risk across studies) is provided in footnotes. The **corresponding risk** (and its 95% confidence interval) is based on the assumed risk in the comparison group and the **relative effect** of the intervention (and its 95% CI).

**CI**: Confidence interval. **Pc**: Proportion in control group with outcome. **RR**: Risk ratio. **DARIS**: Diversity-adjusted required information size. **NA**: Not applicable. **NR**: Not reported.

GRADE Working Group grades of evidence.

**High quality**: Further research is very unlikely to change our confidence in the estimate of effect.

**Moderate quality**: Further research is likely to have an important impact on our confidence in the estimate of effect and may change the estimate.

**Low quality**: Further research is very likely to have an important impact on our confidence in the estimate of effect and is likely to change the estimate.

**Very low quality**: We are very uncertain about the estimate.

mortality with 20% or more. The sensitivity analyses showed that incomplete outcome data bias had the potential to influence the results of all the vaccines (S1 File).

**Network meta-analysis on mortality.** Network meta-analysis on mortality included 7 trials randomizing 168 701 participants comparing placebo, inactivated, mRNA, protein-subunit, and viral vector vaccines. There was no evidence of violation of the transitivity assumption for age and sex (Table 5). The network plot showed a star network without closed loops (S1 File).

All the individual vaccines except NVX-CoV2373-Novavax and Gam-COVID-Vac-Sputnik-V may decrease mortality, but their confidence intervals were compatible with no effect:

**Table 2. Summary of findings (mRNA vaccines).**

mRNA vaccines versus placebo

**Population**: General population
**Settings**: Outpatient
**Intervention**: mRNA vaccine (mRNA-1273-Spikevax and BNT162b2-Corminarty)
**Comparison**: Placebo

| Outcomes | Illustrative comparative risks* (95% CI) | | Relative effect (95% CI) | No of participants (studies) | Quality of the evidence (GRADE) | Comments |
|---|---|---|---|---|---|---|
| | Assumed risk | Corresponding risk | | | | |
| | Control | Vaccine | | | | |
| **All-cause mortality** maximum follow-up | **Study population** | | 0.63 (0.21 to 1.84) | 75 926 (5) | ⊕⊕⊖⊖ **Low** | Downgraded one level for serious risk of bias and one level for serious imprecision. DARIS: 8 409 034 (Pc 0.018%; RRR 20%; alpha 1.67%; beta 10%; diversity 0.0%) |
| | **18 per 100,000** | **12 per 100,000** (4 to 33) | | | | |
| **Vaccine efficacy** Positive test plus symptoms maximum follow-up | **Study population** | | 95% (92 to 97%) | 71 514 (3) | ⊕⊕⊕⊖ **Moderate** | Downgraded one level for serious risk of bias. DARIS: 58 402 (Pc 1.1%; RRR 50%; alpha 1.67%; beta 10%; diversity 0.0%) |
| | **109 per 10,000** | **5 per 10,000** (3 to 9) | | | | |
| **Serious adverse events** maximum follow-up | | | 1.10 (0.91 to 1.33) | 75 926 (5) | ⊕⊕⊖⊖ **Low** | Downgraded one level for serious risk of bias and one level for serious imprecision. DARIS: 211 042 (Pc 0.53%; RRR 20%; alpha 1.67%; beta 10%; diversity 41.4%) |
| | **53 per 10,000** | **58 per 10,000** (48 to 70) | | | | |
| **Health-related quality of life** maximum follow-up | | | NR | NR | NA | No trials assessed health-related quality of life |
| | NR | NR | | | | |
| **Non- serious adverse events** maximum follow-up | **2634 per 10,000** | **4715 per 10,000** (3951 to 5584) | 1.79 (1.50 to 2.12) | 75 898 (5) | ⊕⊕⊕⊖ **Moderate** | Downgraded one level for serious risk of bias. DARIS: 408 514 (Pc 26.3%; RRR 20%; alpha 1.67%; beta 10%; diversity 99.14%) |

*The basis for the **assumed risk** (e.g., the median control group risk across studies) is provided in footnotes. The **corresponding risk** (and its 95% confidence interval) is based on the assumed risk in the comparison group and the **relative effect** of the intervention (and its 95% CI).

**CI**: Confidence interval. **Pc**: Proportion in control group with outcome. **RR**: Risk ratio. **DARIS**: Diversity-adjusted required information size. **NA**: Not applicable. **NR**: Not reported. GRADE Working Group grades of evidence.

**High quality**: Further research is very unlikely to change our confidence in the estimate of effect.

**Moderate quality**: Further research is likely to have an important impact on our confidence in the estimate of effect and may change the estimate.

**Low quality**: Further research is very likely to have an important impact on our confidence in the estimate of effect and is likely to change the estimate.

**Very low quality**: We are very uncertain about the estimate.

Ad26.COV2.S-Janssen (RR, 0.19 [95% CI, 0.01 to 6.65]), ChAdOx1 nCoV-19-Vaxzevria (RR, 0.21 [95% CI, 0.00 to 55.07]), BNT162b2-Corminarty (RR, 0.50 [95% CI 0.00 to 52.70]), CoronaVac (RR, 0.50 [95% CI 0.03 to 7.35]), mRNA-1273-Spikevax (RR, 0.67 [95% CI 0.01 to 37.52]), NVX-CoV2373-Novavax (RR, 1.00 [95% CI 0.00 to 218.18]), and Gam-COVID-Vac-Sputnik-V (RR, 0.99 [95% CI 0.08 to 244.08). CINeMA was very low for all comparisons (S1 File). The between-study variance could not be estimated due to the small number of trials per vaccine comparison.

Based on the surface under the cumulative ranking curves, the Ad26.COV2.S-Janssen vaccine had the greatest likelihood of being the most effective vaccine in reducing mortality (P-

**Table 3. Summary of findings (protein-subunit vaccines).**

Protein-subunit vaccines versus placebo

**Population**: General population
**Settings**: Outpatient
**Intervention**: Protein-subunit (SCB-2019 and NVX-CoV2373-Novavax)
**Comparison**: Placebo

| Outcomes | Illustrative comparative risks* (95% CI) | | Relative effect (95% CI) | No of participants (studies) | Quality of the evidence (GRADE) | Comments |
|---|---|---|---|---|---|---|
| | **Assumed risk** | **Corresponding risk** | | | | |
| | **Control** | **Vaccine** | | | | |
| **All-cause mortality** maximum follow-up | **Study population** | | 0.46 (0.09 to 2.36) | 15 634 | ⊕⊖⊖⊖ **Very low** | Downgraded one level for serious risk of bias and two levels for very serious imprecision. DARIS: 6 901 325 (Pc 0.013%; RRR 20%; alpha 1.67%; beta 10%; diversity 0.0%) |
| | **13 per 100,000** | **6 per 100,000** (1 to 30) | | | | |
| **Vaccine efficacy** Positive test plus symptoms maximum follow-up | **Study population** | | 77% (-5 to 95%) | 17 737 (2) | ⊕⊕⊖⊖ **Low** | Downgraded one level for serious risk of bias and one level for serious imprecision. DARIS: 154 351 (Pc 1.6%; RRR 20%; alpha 1.67%; beta 10%; diversity 93.2%) |
| | **156 per 10,000** | **51 per 10,000** (8 to 164) | | | | |
| **Serious adverse events** maximum follow-up | | | 1.01 (0.66 to 1.55) | 16 389 (4) | ⊕⊕⊖⊖ **Low** | Downgraded one level for serious risk of bias and one level for serious imprecision. DARIS: 241 981 (Pc 0.50%; RRR 20%; alpha 1.67%; beta 10%; diversity 41.4%) |
| | **50 per 10,000** | **51 per 10,000** (33 to 78) | | | | |
| **Health-related quality of life** maximum follow-up | | | NA | NA | NA | No trials assessed health-related quality of life |
| | NA | NA | | | | |
| **Non- serious adverse events** maximum follow-up | **1978 per 1,0000** | **3678 per 10,000** (2413 to 5613) | 1.86 (1.22 to 2.84) | 16 959 (8) | ⊕⊕⊕⊖ **Moderate** | Downgraded one level for serious risk of bias DARIS: 280 894 (Pc 19.8%; RRR 20%; alpha 1.67%; beta 10%; diversity 98.21%) |

*The basis for the **assumed risk** (e.g., the median control group risk across studies) is provided in footnotes. The **corresponding risk** (and its 95% confidence interval) is based on the assumed risk in the comparison group and the **relative effect** of the intervention (and its 95% CI).

**CI**: Confidence interval. **Pc**: Proportion in control group with outcome. **RR**: Risk ratio. **DARIS**: Diversity-adjusted required information size. **NA**: Not applicable. **NR**: Not reported.

GRADE Working Group grades of evidence.

**High quality**: Further research is very unlikely to change our confidence in the estimate of effect.

**Moderate quality**: Further research is likely to have an important impact on our confidence in the estimate of effect and may change the estimate.

**Low quality**: Further research is very likely to have an important impact on our confidence in the estimate of effect and is likely to change the estimate.

**Very low quality**: We are very uncertain about the estimate.

score, 68.6%) followed by ChAdOx1 nCoV-19-Vaxzevria (P-score, 60.9%), CoronaVac (P-score, 50.9%), mRNA-1273-Spikevax (P-score, 48.5%), BNT162b2-Corminarty (P-score, 48.2%), NVX-CoV2373-Novavax (P-score, 42.8%), Gam-COVID-Vac-Sputnik-V (P-score, 43.2%), and placebo (P-score, 37.1%) (Fig 3).

 **Analyses of all vaccines.** When we analyzed all eighteen vaccines that reported on all-cause mortality, meta-analysis (RE) suggested that the vaccines versus placebo result in a large reduction of all-cause mortality (RR, 0.41 [95% CI 0.22 to 0.77]; p = 0.0049; I$^2$ = 0.0%; 212 482

**Table 4. Summary of findings (viral vector vaccines).**

Viral vector vaccines versus control

**Population**: General population
**Settings**: Outpatient
**Intervention**: Viral vector vaccine (Ad26.COV2.S-Janssen, ChAdOx1 nCoV-19-Vaxzevria, and Gam-COVID-Vac-Sputnik-V)
**Comparison**: Control (placebo or MenACWY)

| Outcomes | Illustrative comparative risks* (95% CI) | | Relative effect (95% CI) | No of participants (studies) | Quality of the evidence (GRADE) | Comments |
|---|---|---|---|---|---|---|
| | Assumed risk | Corresponding risk | | | | |
| | Control | Vaccine | | | | |
| **All-cause mortality** maximum follow-up | **Study population** | | 0.25 (0.09 to 0.67) | 67 563 (3) | ⊕⊕⊖⊖ **Low** | Downgraded one level for serious risk of bias and one level for serious imprecision. DARIS: 1 092 776 (Pc 0.067%; RRR 20%; alpha 1.67%; beta 10%; diversity 0.0%) |
| | **67 per 100,000** | **17 per 100,000** (6 to 45) | | | | |
| **Vaccine efficacy** Positive test plus symptoms maximum follow-up | **Study population** | | **69%**, (44% to 83) | 71 702 (5) | ⊕⊕⊕⊖ **Moderate** | Downgraded one level for serious risk of bias DARIS: 122 459 (Pc 1.2%; RRR 50%; alpha 1.67%; beta 10%; diversity 88.35%) |
| | **117 per 10,000** | **36 per 10,000** (20 to 51) | | | | |
| **Serious adverse events** maximum follow-up | | | 0.82 (0.64 to 1.05) | 68 640 (4) | ⊕⊕⊖⊖ **Low** | Downgraded one level for serious risk of bias and one level for serious imprecision. DARIS: 319 505 (Pc 0.42%; RRR 20%; alpha 1.67%; beta 10%; diversity 41.4%) |
| | **42 per 10,000** | **35 per 10,000** (27 to 44) | | | | |
| **Health-related quality of life** maximum follow-up | | | NR | NR | NA | No trials assessed health-related quality of life |
| | NR | NR | | | | |
| **Non- serious adverse events** maximum follow-up | **1854 per 10,000** | **2243 per 10,000** (1854 to 2706) | 1.21 (1.00 to 1.46) | 8 909 (3) | ⊕⊖⊖⊖ **Very low** | Downgraded one level for serious risk of bias and two levels for very serious imprecision DARIS: 32 620 (Pc 18.5%; RRR 20%; alpha 1.67%; beta 10%; diversity 83.3%) |

*The basis for the **assumed risk** (e.g., the median control group risk across studies) is provided in footnotes. The **corresponding risk** (and its 95% confidence interval) is based on the assumed risk in the comparison group and the **relative effect** of the intervention (and its 95% CI).

**CI**: Confidence interval. **Pc**: Proportion in control group with outcome. **RR**: Risk ratio. **DARIS**: Diversity-adjusted required information size. **NA**: Not applicable. **NR**: Not reported.

GRADE Working Group grades of evidence.

**High quality**: Further research is very unlikely to change our confidence in the estimate of effect.

**Moderate quality**: Further research is likely to have an important impact on our confidence in the estimate of effect and may change the estimate.

**Low quality**: Further research is very likely to have an important impact on our confidence in the estimate of effect and is likely to change the estimate.

**Very low quality**: We are very uncertain about the estimate.

participants; low certainty; S1 File). TSA, however, showed that we did not have enough data to confirm that the vaccines reduced mortality with 20% or more (S1 File).

## Prevention of symptomatic COVID-19 participants with positive PCR test

**Inactivated vaccines.** Three trials assessing inactivated vaccines reported vaccine efficacy on symptomatic COVID-19 participants. Meta-analysis (FE) showed that inactivated vaccines versus placebo result in a large reduction of symptomatic COVID-19 participants (efficacy,

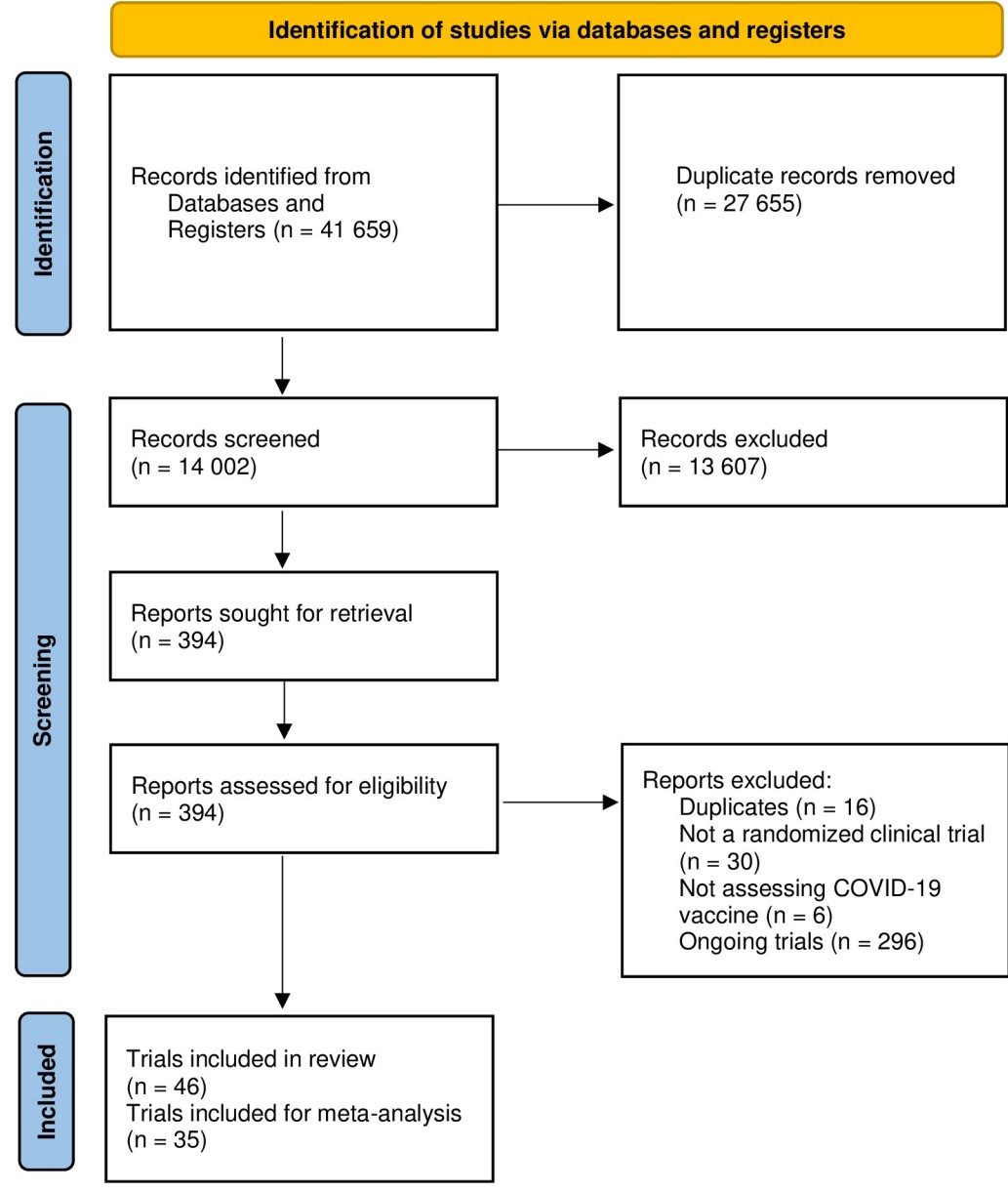

**Fig 1. PRISMA flowchart.**

61% [95% CI 52% to 68%]; p<0.0001; $I^2$ = 80.5%; 48 029 participants; moderate certainty; Fig 4). Visual inspection of the forest plot and $I^2$ suggested evidence of substantial heterogeneity.

**mRNA vaccines.** Three trials assessing mRNA vaccines reported vaccine efficacy on symptomatic COVID-19 participants. Meta-analysis (RE) showed that mRNA vaccines versus placebo result in a large reduction of symptomatic COVID-19 participants (efficacy, 95% [95% CI 92% to 97%]; p<0.0001; $I^2$ = 0.0%; 71 514 participants; moderate certainty; Fig 4).

**Protein-subunit vaccines.** Two trials assessing protein-subunit vaccines reported vaccine efficacy on symptomatic COVID-19 participants. Meta-analysis (RE) showed that protein-sub-unit vaccines versus placebo may result in a large reduction of symptomatic COVID-19 partici-pants, but the confidence interval was compatible with no effect (efficacy, 77% [95% CI −5%

**Table 5. Randomized clinical trials included in systematic review of vaccines against COVID-19.**

| Source–First author and year of publication | Trial registration | Trial phase | Vaccine name [1] | Developer/Investigator | Number randomized [2] | Overall risk of bias [2] |
|---|---|---|---|---|---|---|
| | | | mRNA Vaccines | | | |
| Baden et al, 2020 | NCT04470427 | 3 | mRNA-1274/ Spikevax | Moderna, Inc | 30420 | Some concerns |
| Walsh et al, 2020 18-55y 30μg | NCT04368728 | 1 | BNT162b2/ Comirnaty | BioNTech SE, Pfizer Inc. | 195 | Some concerns |
| Walsh et al, 2020 65-85y 30μg | | | | | | Some concerns |
| Mulligan et al, 2020 | NCT04368728 | 1/2 | BNT162b1 | | 45 | Some concerns |
| Polack et al, 2020 | NCT04368728 | 3 | BNT162b2/ Comirnaty | | 43548 | Some concerns |
| Li et al, 2021 | ChiCTR2000034825, NCT04523571 | 1 | BNT162b1 | BioNTech, Shanghai Fosun Pharmaceutical Development | 144 | |
| | | | Viral Vector Vaccines | | | |
| Madhi et al, 2021 | NCT04444674, PACTR202006922165132 | 1/2 | ChAdOx1 nCoV-19/ AZD 1222/Vaxzevria | University of Oxford, AstraZeneca | 2026 | Low risk of bias |
| Ramasamy et al, 2020 | NCT04400838, ISRCTN15281137 | 2/3 | ChAdOx1 nCoV-19/ AZD 1222/Vaxzevria | | 560 | Some concerns |
| Folegatti et al, 2020 | NCT04324606 | 3 | ChAdOx1 nCoV-19/ AZD 1222/Vaxzevria | | 1067 | Low risk of bias |
| Voysey et al, 2020 UK | NCT04400838, ISRCTN89951424 | 3 | ChAdOx1 nCoV-19/ AZD 1222/Vaxzevria | | 20675 (received all doses of intervention) | High risk of bias |
| Voysey et al, 2020 Brazil | | | | | | High risk of bias |
| Logunov et al, 2021 | NCT04530396 | 3 | Gam-COVID-Vac/ Sputnik V | Gamaleya Research Institute of Epidemiology and Microbiology | 21977 | Some concerns |
| Stephenson et al, 2021 | NCT04505722 | 1 | Ad26.COV2.S/ Janssen COVID-19 Vaccine/JNJ-78436735/Ad26COVS1 | Janssen Pharmaceuticals | 25 | |
| Sadoff et al, 2020 | NCT04436276 | 1/2 | Ad26.COV2.S/ Janssen COVID-19 Vaccine/JNJ-78436735/Ad26COVS1 | | 402 | Low risk of bas |
| Sadoff et al, 2021 | NCT04505722 | 3 | Ad26.COV2.S/ Janssen COVID-19 Vaccine/JNJ-78436735/Ad26COVS1 | | 43783 | Some concerns |
| Cai Zhu et al, 2020 | NCT04341389, NCT04341390 | 2 | Ad5-nCOV/ Convidicea | CanSinoBIO | 508 | Some concerns |
| | | | Protein Subunit Vaccines | | | |
| Keech et al, 2020 | NCT04368988 | 1/2 | NVX-CoV2373 | Novavax Inc. | 125 | Some concerns |
| Shinde et al, 2021 | NCT04533399 | 1/2 | NVX-CoV2373 | | 4406 | High risk of bias |
| Formica et al, 2021 | NCT04368988 | 1/2 | NVX-CoV2373 | | 1288 | Some concerns |
| Heath et al, 2021 | NCT04583995 | 3 | NVX-CoV2374 | | 15187 | |
| Richmond et al, 2021 18-54y | NCT04405908 | 1 | SCB-2019 | Clover Biopharmaceuticals, Coalition for Epidemic Preparedness Innovations | 151 | Some concerns |
| Richmond et al, 2021 55-75y | | | | | | Some concerns |
| Goepfert et al, 2021 | NCT04537208 | 1/2 | CoV2 preS dTM | Sanofi Pasteur, GlaxoSmithKline | 441 | Some concerns |

*(Continued)*

**Table 5.** (Continued)

| Source–First author and year of publication | Trial registration | Trial phase | Vaccine name [1] | Developer/Investigator | Number randomized [2] | Overall risk of bias [2] |
|---|---|---|---|---|---|---|
| Yang et al, 2021 phase 1 | NCT04445194 | 1 | ZF2001 | Anhui Zhifei Longcom Biopharmaceutical Co., Ltd, Institute of Microbiology—Chinese Academy of Sciences | 50 | Some concerns |
| Yang et al, 2021 phase 2 | NCT04466085 | 2 | | | 900 | Some concerns |
| Ward et al, 2020 | NCT04450004 | 1 | CoVLP | Medicago Inc. | 180 | |
| Gobeil et al, 2021 | NCT04636697 | 2 | CoVLP | | 588 | Some concerns |
| Chappell et al, 2021 | NCT04495933 | 1 | SARS-CoV-2 sclamp vaccine | The University of Queensland, Syneos Health, CSIRO Manufacturing, Seqirus, Coalition for Epidemic Preparedness Innovations | 120 | |
| Inactivated Virus Vaccines | | | | | | |
| Ella et al, 2021 | NCT04471519 | 1 | BBV152/Covaxin | Bharat Biotech | 375 | Some concerns |
| Wu et al 2021 phase 1 | NCT04383574 | 1 | CoronaVac | Sinovac Biotech | 422 | Low Risk of Bias |
| Wu et al, 2021 phase 2 | | 2 | | | | Low Risk of Bias |
| Zhang et al, 2020 | NCT04352608 | 1/2 | CoronaVac | | 744 | Some concerns |
| Han et al, 2021 | NCT04551547 | 1/2 | CoronaVac | | 72 phase 1 480 phase 2 | |
| Bueno et al, 2021 | NCT04651790 | 3 | CoronaVac | | 434 | High risk of bias |
| Xia et al, 2020 phase 1 | ChiCTR2000031809 | 1 | BBIBP-CorV/ Vero cell | Bejing Institute of Biological Products/ Sinopharm (CNBG) | 320 | Low Risk of Bias |
| Xia et al, 2020 phase 2 | | 2 | | | | Low Risk of Bias |
| Pu et al, 2020 | CTR20200943, NCT04412538 | 1 | SARS-CoV-2 inactivated vaccine | Institute of Medical Biology, Chinese Academy of Medicine Science | 192 | High risk of bias |
| Che et al, 2020 | NCT04412538 | 2 | SARS-CoV-2 inactivated vaccine | | 750 | Some concerns |
| Palacios et al, 2021 | NCT04456595 | 3 | CoronaVac | Fundação Butantan and São Paulo Research Foundation | 12408 | Low Risk of Bias |
| Pan et al, 2021 | ChiCTR2000038804, ChiCTR2000039462 | 1/2 | KCONVAC | Shenzhen Kangtai Biological Products Co. Ltd, Beijing Minhai Biotechnology | 60 phase 1 500 phase 2 | Some concerns |
| Al Kaabi et al, 2021 | NCT04510207; ChiCTR2000034780 | 3 | WIV04, HB02 | Sinopharm China National Biotec Group Company Limited, Wuhan Institute of Biological Products Co. Ltd., the Beijing Institute of Biological Products Co. Ltd | 40411 | Low risk of bias |

[1] If multiple names were used at different stages of development, all of them are listed.

[2] If not number randomized it is noted differently.

to 95%]; p = 0.06; $I^2$ = 92.6%; 17 737 participants; low certainty; Fig 4). Visual inspection of the forest plot and $I^2$ suggested substantial evidence of heterogeneity.

**Viral vector vaccines.** Five trials assessed viral vector reported vaccine efficacy on symptomatic COVID-19 participants. Meta-analysis (RE) showed that viral vector vaccines versus placebo likely result in a large reduction of symptomatic COVID-19 participants (efficacy,

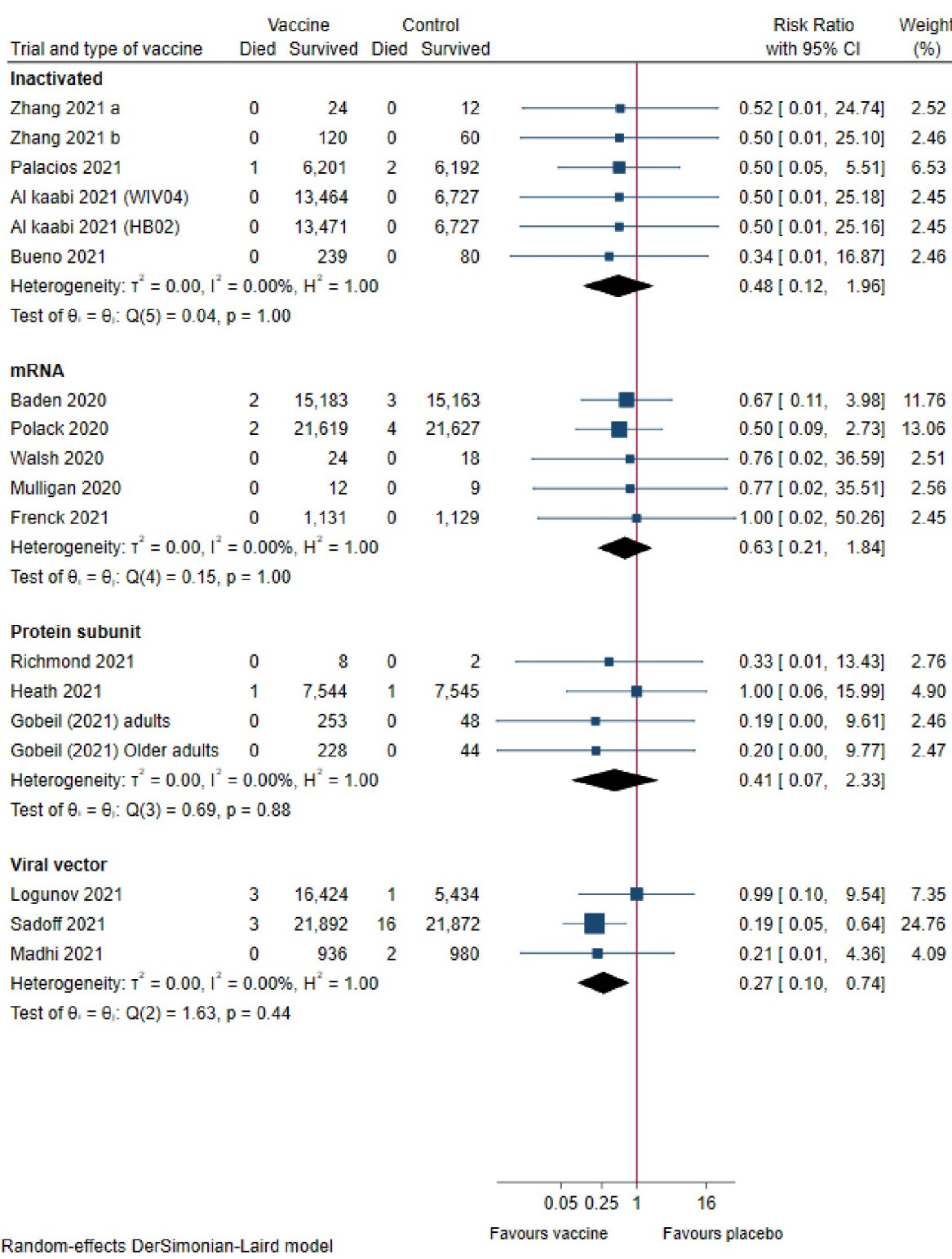

Fig 2. COVID-19 vaccines versus placebo on all-cause mortality.

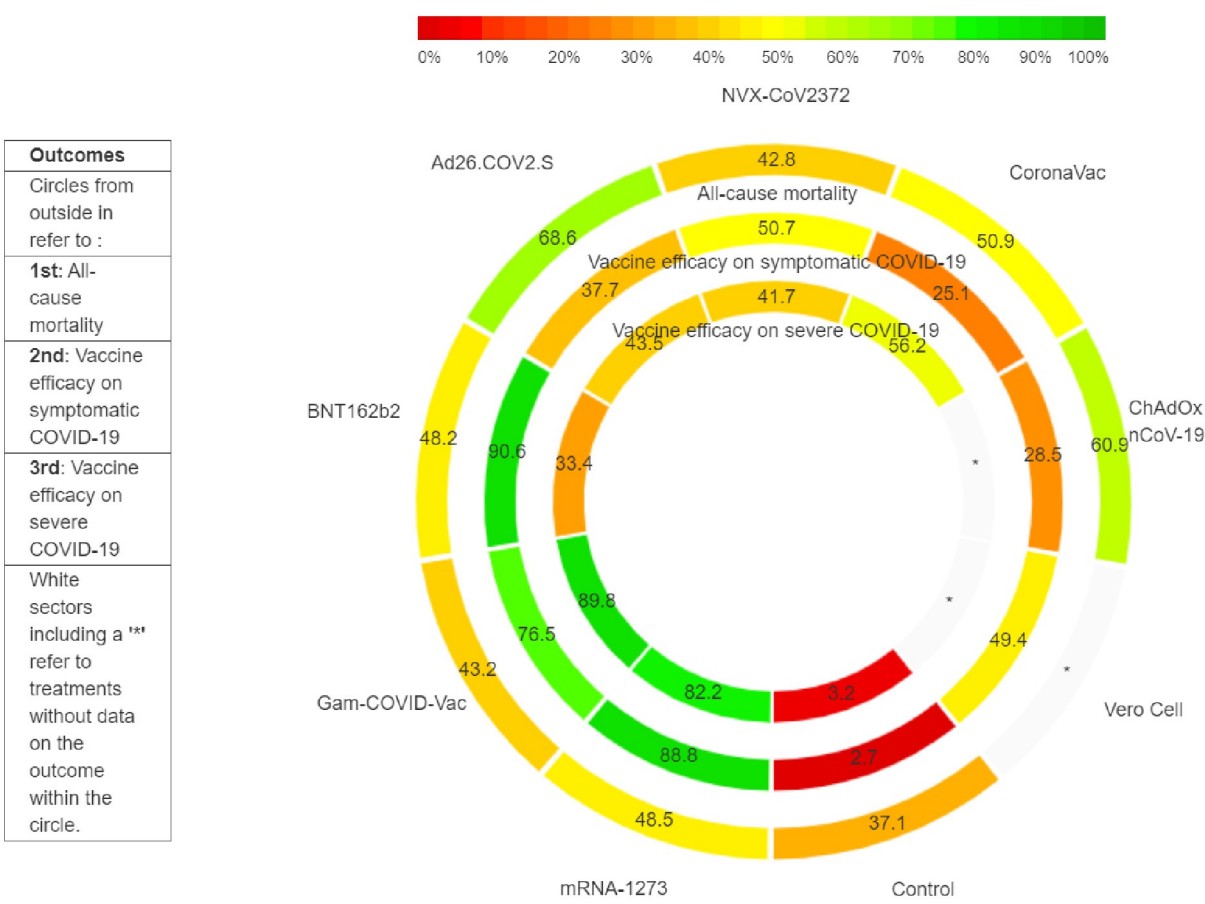

**Fig 3. Rank heat plot for all-mortality and vaccine efficacy on symptomatic COVID-19 and on severe COVID-19.**

69% [95% CI 44% to 83%]; p = 0.0001; $I^2$ = 85.7%; 70 865 participants; moderate certainty; Fig 4). Visual inspection of the forest plot, $I^2$, and estimation of between-study variance ($\tau^2$ = 0.38) indicated substantial heterogeneity.

Subgroup analyses of the different vaccines (favoring the Gam-COVID-Vac-Sputnik-V vaccine) and regarding vested interests (favoring without vested interests) showed evidence of differences (p<0.05) (S1 File).

**Network meta-analysis on prevention of symptomatic COVID-19 participants with positive PCR test.** Network meta-analysis on vaccine efficacy included eleven trials randomizing 205 916 participants comparing placebo/control with inactivated, mRNA, protein-subunit, or viral vector vaccines. There was no evidence of violation of the transitivity assumption for age and sex (Table 5).

All vaccines were more effective than placebo. The two mRNA vaccines (mRNA-1273-Spikevax and BNT162b2-Corminarty) and the viral vector vaccine, Gam-COVID-Vac-Sputnik-V, were likely superior to the remaining vaccines. Three vaccines achieved a vaccine efficacy superior to the minimum clinical important effectiveness of 50% compared with placebo: BNT162b2-Corminarty (efficacy, 95% [95% CI 83% to 99%]), mRNA-1273-Spikevax (efficacy, 95% [95% CI 80% to 99%]), and Gam-COVID-Vac-Sputnik-V (efficacy, 90% [95% CI 63% to 97%]). Five vaccines did not achieve a vaccine efficacy which was significantly superior to

**Fig 4. COVID-19 vaccine efficacy on preventing symptomatic COVID-19 participants with positive PCR.**

50%: NVX-CoV2373-Novavax (efficacy, 77% [95% CI 39% to 91%]), Vero Cell (efficacy, 75% [95% CI 38% to 90%]), Ad26.COV2.S-Janssen (efficacy, 66% [95% CI −17% to 90%]), Corona-Vac (efficacy, 50% [95% CI −73% to 86%]), and ChAdOx1 nCoV-19-Vaxzevria (efficacy, 56% [95% CI 4% to 80%]). CINeMA was very low for all comparisons (S1 File).

Based on the surface under the cumulative ranking curves, the BNT162b2-Corminarty vaccine had the greatest likelihood of being the most effective vaccine in reducing symptomatic COVID-19 (P-score, 90.6%) (Fig 3). It was followed by mRNA-1273-Spikevax (P-score, 88.8%), Gam-COVID-Vac-Sputnik-V (P-score, 76.5%), NVX-CoV2373-Novavax (P-score, 50.7%), Vero Cell (P-score, 49.4%), Ad26.COV2.S-Janssen (P-score, 37.7%), ChAdOx1 nCoV-19-Vaxzevria (P-score, 28.5%), CoronaVac (P-score, 25.1%), and placebo (P-score, 2.7%).

**Prevention of severe COVID-19 symptoms with positive PCR test or prevention of positive PCR test only.** Vaccine efficacies on prevention of asymptomatic and severe disease are presented in the supplementary material (S1 File).

**Serious adverse events.** *Inactivated vaccines.* Seven trials assessing inactivated vaccines reported on serious adverse events. Meta-analysis (RE) showed that inactivated vaccines versus placebo may decrease serious adverse events slightly, but the confidence interval was also compatible with no effect (RR, 0.84 [95% CI 0.68 to 1.06]; p = 0.15; $I^2$ = 0.0%; 53 839 participants; low certainty; S1 File).

*mRNA vaccines.* Five trials assessing mRNA vaccines reported on serious adverse events. Meta-analysis (RE) showed that mRNA vaccines versus placebo may increase serious adverse events slightly, but the confidence interval was also compatible with no effect (RR, 1.10 [95% CI 0.91 to 1.33]; p = 0.31; $I^2$ = 0.0%; 75 926 participants; low certainty; S1 File).

*Protein-subunit vaccines.* Four trials assessing protein-subunit vaccines reported on serious adverse events. Meta-analysis (RE) showed that these vaccines versus placebo have little or no effect on serious adverse events (RR, 1.01 [95% CI 0.66 to 1.55]; p = 0.97; $I^2$ = 0.0%; 16 389 participants; low certainty; S1 File).

*Viral vector vaccines.* Four trials assessing viral vector reported on serious adverse events. Meta-analysis (FE) showed that viral vector vaccines versus control may decrease serious adverse events, but the confidence interval was also compatible with no effect (RR, 0.82 [95% CI 0.64 to 1.05]; p = 0.12; $I^2$ = 0.0%; 68 640 participants; low certainty; S1 File).

*Health-related quality of life.* None of the included trials assessed health-related quality of life.

**Non-serious adverse events.** *Inactivated vaccines.* Eleven trials assessing inactivated vaccines reported on adverse events considered non-serious. Meta-analysis (RE) showed that inactivated vaccines versus placebo have little or no effect on adverse events not considered serious (RR, 1.02 [95% CI 0.92 to 1.13]; p = 0.67; $I^2$ = 92.4%; 54 239 participants; moderate certainty; S1 File).

*mRNA vaccines.* Five trials assessing mRNA vaccines reported on adverse events considered non-serious. Meta-analysis (RE) showed that mRNA vaccines versus placebo likely result in a large increase of adverse events not considered serious (RR, 1.79 [95% CI 1.50 to 2.12]; p<0.0001; $I^2$ = 94.7%; 75 898 participants; moderate certainty; S1 File).

*Protein-subunit vaccines.* Seven trials assessing protein-subunit vaccines reported on adverse events considered non-serious. Meta-analysis (RE) showed that protein-subunit vaccines versus placebo result in a large increase of adverse events not considered serious (RR, 1.86 [95% CI 1.22 to 2.84]; p = 0.004; $I^2$ = 87.2%; 16 959 participants; moderate certainty; S1 File).

*Viral vector vaccines.* Three trials assessing viral vector vaccines reported on adverse events considered non-serious. Meta-analysis (RE) showed that viral vector vaccines versus control may increase of adverse events not considered serious, but the confidence interval was also compatible with no effect (RR, 1.21 [95% CI 1.00 to 1.85]; p = 0.05; $I^2$ = 80.9%; 8 909 participants; very low certainty; S1 File).

## Discussion

The assessed vaccines (Ad26.COV2.S-Janssen, BNT162b2-Corminarty, ChAdOx1 nCoV-19-Vaxzevria, CoronaVac, Gam-COVID-Vac-Sputnik-V, mRNA-1273-Spikevax,

NVX-CoV2373-Novavax, and Vero Cell) all seem to be effective in preventing symptomatic COVID-19. Network meta-analysis suggests that the mRNA vaccines (mRNA-1273-Spikevax and BNT162b2-Corminarty) and the viral vector vaccine, Gam-COVID-Vac-Sputnik-V, could be superior to the remaining vaccines in preventing symptomatic COVID-19.

Our pairwise meta-analysis showed that the viral vector vaccines decreased mortality. Network meta-analysis suggested that the viral vector vaccine Ad26.COV2.S-Janssen was most effective in reducing mortality. All COVID-19 vaccines were more likely to reduce mortality than placebo or control, but longer follow up and more participants are needed to confirm this with a higher degree of certainty.

All vaccines, except inactivated vaccines, increased the risk of having a non-serious adverse event. We found no difference on serious adverse events, for any of the included vaccines (Ad26.COV2.S-Janssen, BNT162b2-Corminarty, CoronaVac, Gam-COVID-Vac-Sputnik-V, mRNA-1273-Spikevax, NVX-CoV2373-Novavax, and Vero Cell), but our Trial Sequential Analyses indicated that we did not have enough data to reject any differences.

Our results should been seen together with observational evidence suggesting rare serious adverse events associated with the vaccines [68]. The results of our review is supported by the first nationwide observational studies showing high vaccine efficacy of BNT162b2-Corminarty and of BNT162b2-Corminarty and ChAdOx1 nCoV-19-Vaxzevria against COVID-19 hospital admissions [69, 70].

We included 35 trials randomizing a total of 219 864 participants that contributed to our analyses. Of these trials, ten were at overall low risk of bias, seventeen at overall some concerns, and eight trials at overall high risk of bias. The certainty of evidence according to GRADE ranged from very low to moderate for the pairwise meta-analyses and the CINeMA were very low for all our network meta-analyses.

Most trials compared vaccines versus placebo. A few trials compared the vaccine versus a control vaccine not active towards SARS-CoV2 (MenACWY) [37, 40, 45]. Only trials assessing inactivated, mRNA, protein-subunit, and recombinant viral vector vaccines contributed to the assessment of our primary outcomes.

We found very similar results for efficacy of vaccine prevention for the two mRNA-based vaccines. This is not surprising as the construction differs only in details of the formulation of nanoparticles used for protection of the mRNA. In contrast, marked heterogeneity was observed regarding the three adenovector-based vaccines. Again, this is perhaps not surprising as the adenoviral backbones are different: ChAdOx1 nCoV-19, Ad26, or Ad5. Preclinical studies in mice and non-human primates have pointed to significant variation in the capacity of different adenoviral backbones to induce adaptive immunity [71–73]. Another cause could be the fact that the vaccines were not compared head-to-head in the same populations under similar conditions.

The emergence of SARS-CoV-2 variants of concern like in the UK (alpha/B.1.1.7), South Africa (B.1.351/Beta), India (B.1617.1/Kappa and B.1.617.2/Delta), Brazil (B.1.1.284/Gamma), USA (B.1.427 and B.1.429 / Epsilon), and Peru (C.37/Lambda) draws attention to increased transmissibility, higher disease severity, and evasion of the immune system through mutations in the spike protein [74–77]. This should be taken into consideration when trying to compare vaccine efficacies derived from studies carried out at different time-points and/or locations. The efficacy of existing vaccines against emerging variants needs to be closely monitored and the adaptability of the vaccines to potential immunity evading mutations needs to be assessed.

The trials in our systematic review did not include pregnant participants. Pregnant women constitute a vulnerable group for COVID-19 [78, 79]. Systematic reviews of observational studies showed that compared to pregnant women without COVID-19, pregnant women with this infection may require more frequent admissions to intensive care units (odds ratio (OR),

1.62 [95% CI 1.33 to 1.96]) and invasive ventilation (OR,1.88 [95% CI 1.36 to 2.60]) [78, 79]. For these reasons, many countries like the USA [80–82] and Israel [83] are presently assessing vaccine effectiveness in pregnant women. The World Health Organization (WHO) recommended mRNA vaccines only to pregnant women at highest risk after consultation with their physician [84], and trials are ongoing [85, 86].

Our review has several strengths. Our methodology was described in detail in a protocol published before the literature searches were initiated [8]. We systematically assessed the risks of systematic errors through bias risk assessments, we conducted Trial Sequential Analyses to control random errors and guide our GRADE assessments of the domain 'imprecision', and we adjusted our thresholds for statistical significance to control the risks of random errors [15]. Our review is to our knowledge the first review to perform network meta-analysis comparing the different COVID-19 vaccines. It is also more updated than previous systematic reviews [7, 87, 88]. Our living systematic review will be continually updated to incorporate relevant new evidence as it becomes available [8, 89, 90].

Our review also has limitations. Most of the larger phase III trials presented only interim analyses. The follow-up ranged from only 35 to 92 days. A longer follow-up would most likely yield more events and lead to more robust conclusions. Moreover, the trials did not report vaccine efficacy similarly. They used different cut-offs, follow-up durations, and definitions of vaccine efficacy. Especially the efficacy of preventing severe COVID-19 was not equally reported for the different vaccines. Furthermore, most trials only reported efficacy in the group of participants followed more than two weeks after the last vaccination and only reported adverse events for a confined time. Accordingly, we may look at biased results. Moreover, reporting of adverse events was variably and insufficient in several trials.

The trials were conducted in different countries. The participants of the different trials could therefore have been exposed to different variants of SARS-CoV2. We were unable to fully extract comparable data from the ChAdOx1 nCoV-19-Vaxzevria trials due to the lack relevant data from the individual trials, as the trials reported their results in a meta-analysis [37, 45, 52, 91].

## Conclusion

Our systematic review shows that the inactivated vaccines, mRNA vaccines, protein-subunit vaccines, and viral vector vaccines are effective in preventing infection with SARS-CoV2. Current evidence shows that mRNA vaccines seem most effective in preventing COVID-19, but viral vector vaccines seem most effective in reducing mortality. The inactivated vaccine (CoronaVac), mRNA vaccines (mRNA-1273-Spikevax and BNT162b2-Corminarty), protein-sub-unit vaccine (NVX-CoV2373-Novavax), and viral vector vaccines (Gam-COVID-Vac-Sputnik-V, ChAdOx1 nCoV-19-Vaxzevria, and Ad26.COV2.S- Janssen) do not seem to increase the risk of serious adverse events according to published randomized trials. Most of the vaccines increase the risk of adverse events not considered serious. Our results may help guide authorities when deciding what vaccines to incorporate into their vaccine programs weighing our results against the potential rare events seen in observational studies.

## Supporting information

**S1 File.**
(DOCX)

**S1 Checklist.**
(DOCX)

## Acknowledgments

The living systematic review was supported by The Copenhagen Trial Unit, Centre for Clinical Intervention Research, The Capital Region, Copenhagen University Hospital–Rigshospitalet, Copenhagen, Denmark.

## Author Contributions

**Conceptualization:** Steven Kwasi Korang, Christian Gluud.

**Data curation:** Steven Kwasi Korang.

**Formal analysis:** Steven Kwasi Korang, Areti Angeliki Veroniki, Christian Gluud.

**Investigation:** Steven Kwasi Korang, Elena von Rohden, Giok Ong, Owen Ngalamika, Faiza Siddiqui, Sophie Juul, Emil Eik Nielsen, Joshua Buron Feinberg, Johanne Juul Petersen, Christian Legart, Mathias Maagaard, Sarah Klingenberg, Ariel Bardach, Agustín Ciapponi.

**Methodology:** Steven Kwasi Korang, Areti Angeliki Veroniki, Lehana Thabane, Ariel Bardach, Agustín Ciapponi, Allan Randrup Thomsen, Janus C. Jakobsen, Christian Gluud.

**Project administration:** Steven Kwasi Korang, Elena von Rohden, Christian Gluud.

**Resources:** Sarah Klingenberg.

**Software:** Areti Angeliki Veroniki.

**Supervision:** Christian Gluud.

**Writing – original draft:** Steven Kwasi Korang, Elena von Rohden, Afoke Kokogho, Ariel Bardach, Agustín Ciapponi, Allan Randrup Thomsen, Christian Gluud.

**Writing – review & editing:** Steven Kwasi Korang, Elena von Rohden, Areti Angeliki Veroniki, Giok Ong, Owen Ngalamika, Faiza Siddiqui, Sophie Juul, Emil Eik Nielsen, Joshua Buron Feinberg, Johanne Juul Petersen, Christian Legart, Afoke Kokogho, Mathias Maagaard, Sarah Klingenberg, Lehana Thabane, Ariel Bardach, Agustín Ciapponi, Allan Randrup Thomsen, Janus C. Jakobsen, Christian Gluud.

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
