## [Decision Letter · Decision Letter 0]

16 Nov 2021

Vaccines to prevent COVID-19 a living systematic review with Trial Sequential Analysis and network meta-analysis of randomized clinical trials

PONE-D-21-31660

Dear Dr. Korang,

We’re pleased to inform you that your manuscript has been judged scientifically suitable for publication and will be formally accepted for publication once it meets all outstanding technical requirements.

Kind regards,

Stefanos Bonovas, M.D., M.Sc., Ph.D.

Academic Editor

PLOS ONE

Additional Editor Comments (optional):

Reviewers' comments:

Reviewer's Responses to Questions

**Comments to the Author**

1. Is the manuscript technically sound, and do the data support the conclusions?

Reviewer #1: Yes

2. Has the statistical analysis been performed appropriately and rigorously? 

Reviewer #1: Yes

3. Have the authors made all data underlying the findings in their manuscript fully available?

Reviewer #1: Yes

4. Is the manuscript presented in an intelligible fashion and written in standard English?

Reviewer #1: Yes

5. Review Comments to the Author

Reviewer #1: This is a living systematic review and network meta-analysis focusing on the available vaccines to prevent COVID-19. In a moment where research is under the magnifying glass, this work becomes fundamental. Written effectively following the best methodology and highlighting important results, I thank and congratulate the authors for the continuous update of the evidence.

6. PLOS authors have the option to publish the peer review history of their article (what does this mean?). If published, this will include your full peer review and any attached files.

Reviewer #1: No

---

## [Editor Report · Acceptance letter]

10 Dec 2021

PONE-D-21-31660 

Vaccines to prevent COVID-19: a living systematic review with Trial Sequential Analysis and network meta-analysis of randomized clinical trials 

Dear Dr. Korang:

I'm pleased to inform you that your manuscript has been deemed suitable for publication in PLOS ONE. Congratulations! Your manuscript is now with our production department. 

Kind regards, 

on behalf of

Dr. Stefanos Bonovas 

Academic Editor

PLOS ONE